# Inhibition of casein kinase 2 induces cell death in tyrosine kinase inhibitor resistant chronic myelogenous leukemia cells

Ondřej Mitrovský[1,2], Denisa Myslivcová[1], Tereza Macháčková-Lopotová[1], Adam Obr[1]*, Kamila Čermáková[3], Šárka Ransdorfová[4], Jana Březinová[4], Hana Klamová[5], Markéta Žáčková[1]

1 Department of Proteomics, Institute of Hematology and Blood Transfusion, Prague 2, Czech Republic, 2 Faculty of Science, Charles University, Prague, Czech Republic, 3 Laboratory of PCR Diagnostics of Leukemias, Institute of Hematology and Blood Transfusion, Prague 2, Czech Republic, 4 Department of Cytogenetics, Institute of Hematology and Blood Transfusion, Prague 2, Czech Republic, 5 Clinical Division, Institute of Hematology and Blood Transfusion, Prague 2, Czech Republic

* adam.obr@uhkt.cz

**Data Availability Statement:** All relevant data are within the manuscript and its Supporting Information files.

## Abstract

Chronic myelogenous leukemia (CML) is a myeloproliferative disease characterized by the BCR-ABL oncogene. Despite the high performance of treatment with tyrosine kinase inhibitors (TKI), about 30% of patients develop resistance to the therapy. To improve the outcomes, identification of new targets of treatment is needed. Here, we explored the Casein Kinase 2 (CK2) as a potential target for CML therapy. Previously, we detected increased phosphorylation of HSP90β Serine 226 in patients non-responding to TKIs imatinib and dasatinib. This site is known to be phosphorylated by CK2, which was also linked to CML resistance to imatinib. In the present work, we established six novel imatinib- and dasatinib-resistant CML cell lines, all of which had increased CK2 activation. A CK2 inhibitor, CX-4945, induced cell death of CML cells in both parental and resistant cell lines. In some cases, CK2 inhibition also potentiated the effects of TKI on the cell metabolic activity. No effects of CK2 inhibition were observed in normal mononuclear blood cells from healthy donors and BCR-ABL negative HL60 cell line. Our data indicate that CK2 kinase supports CML cell viability even in cells with different mechanisms of resistance to TKI, and thus represents a potential target for treatment.

## Introduction

Chronic myeloid leukemia (CML) is a myeloproliferative disease characterized by the presence of Philadelphia translocation t(9;22)(q34;q11), resulting in the expression of a constitutively active BCR-ABL kinase. BCR-ABL is the main driver of CML [1]. A selective BCR-ABL inhibitor, imatinib, has been introduced into clinical practice in 2001 and represents a breakthrough in CML therapy [2,3]. About one-third of patients, however, develop resistance or intolerance to the drug. To address this issue, second- and third-generation tyrosine kinase inhibitors (TKIs)—like dasatinib, nilotinib, and ponatinib—were developed, and are effective against most imatinib-resistant cases of CML.

**Funding:** Research was supported by the Ministry of Health, Czech Republic, (project for conceptual development of research organization No 00023736). The funders had no role in study design, data collection and analysis, decision to publish, or preparation of the manuscript.

**Competing interests:** The authors have declared that no competing interests exist.

Patients treated with advanced generations of TKIs, however, also have a high risk of acquiring resistance to the new drugs [4–6]. Therefore, other ways of targeting resistant CML cells are explored. The research in this area focuses on strategies to target pathways that are not induced by BCR-ABL, including those currently considered as non-oncogenic [7,8].

Casein kinase 2 (CK2) is a ubiquitously expressed serine/threonine (S/T) kinase. It is usually present as a tetrameric complex of two catalytic (α, α') and two regulatory (β) subunits. Although it was previously labeled as constitutively active, several reports now suggest many possible pathways of CK2 regulation, e.g. via protein interactions, differential phosphorylation, and interactions with other regulatory elements [9]. CK2 has a net pro-survival, anti-apoptotic role [10–13]. Its expression is increased in cancer cells [14–16], and its signaling feeds the usual cancer-related pathways, even reaching the point where cancer cells become CK2-dependent [17].

Furthermore, CK2 has also been associated with multi-drug resistance to chemotherapy [18,19]. This all shows CK2 as an important player in cancer pathogenesis, and since some highly specific inhibitors are available, it also represents a possible target for treatment [9]. An orally available compound Silmitasertib (CX-4945) is currently evaluated in clinical trials for several cancer types [20,21], but so far has not been tested for CML treatment [22].

We used protein-antibody arrays to find proteins differentially expressed and/or phosphorylated in samples of patients divided into groups according to their therapy response (as characterized by ELN [23]), and healthy donors. Among others, we detected increased phosphorylation of HSP90β on Ser226 in imatinib- and dasatinib- resistant patients. Phosphorylation of this site was previously identified as CK2-dependent [24,25]. As CK2 is associated with resistance to imatinib [26,27], we further explored the role of CK2 in the resistance to both imatinib and dasatinib. In TKI-resistant cells, CK2 expression was increased, and all the resistants were also highly sensitive to CK2 inhibition. Additionally, our preliminary data from patients non-responding to the therapy showed corresponding effects of CK2 inhibition on cell viability.

## Results

### CK2-dependent phosphorylation of HSP90β is increased in TKI non-responding CML patients

We used protein-antibody arrays to analyze 14 samples of total leukocytes obtained from CML patients with varying responses to standard therapy (S1 Fig in S1 File, Table 1A). We found several differentially phosphorylated proteins compared to healthy donors, including changes at phosphorylation sites of heat-shock proteins HSP90 and HSP27 (S1 Fig in S1 File). In particular, a systematic increase in HSP90β S226 phosphorylation was identified in samples from patients who did not respond to therapy or relapsed, in comparison to healthy donors (S1A Fig in S1 File). For two other patients, we performed the screen at the time of diagnosis and subsequently 10 and 11 months after treatment initiation (Table 1B). The non-responding patient had a markedly higher ratio of HSP90β phosphorylated on S226 to total HSP90β.

### Characterization of six novel, CML-derived, TKI-resistant cell lines

We generated six novel cell lines derived from CML that exhibit resistance to TKIs. JURL-MK1, MOLM-7, and K562 cells which were able to grow in media containing 2 μM imatinib or 2 nM dasatinib, respectively, are further designated as "TKI resistant" (imatinib resistant–IR, dasatinib resistant–DR). As controls, BCR-ABL negative cell lines HL-60 and OCI-AML3 were cultivated under the same conditions as CML cell lines. Although these cells

**Table 1. A. Patient characteristics at the time of analysis.** HD–healthy donors, OR–optimal response to treatment, NR–nonresponding patients, REL–relapsed patients. DG–diagnosis, TH–therapy (imatinib/dasatinib/nilotinib), BCR-ABL (%)–mRNA level, WBC–white blood cell count, PLT–platelet count, BL—% of blast cells in total cell count, mut–mutations in the BCR-ABL kinase domain, ab–karyotype aberations. **B. Patient characteristics at the time of analysis.** NR–nonresponding patient, OR–optimal response to treatment, DG–diagnosis, BCR-ABL (%)–mRNA level, WBC–white blood cell count, PLT–platelet count, BL—% of blast cells in total cell count, TH–therapy.

| Tab 1A | | F/M | months from DG | TH (I/D/N) | BCR-ABL (%) | WBC (10^9/l) | PLT (10^9/l) | BL (%) | mut (n) | ab (n) |
|---|---|---|---|---|---|---|---|---|---|---|
| OR (n = 4) | | 4/0 | 70,3 (12–141) | 3/0/1 | 0 (0–0,001) | 5,35 (3,98–14,4) | 202 (170–268) | 0 | 0 | 0 |
| NR (n = 5) | | 3/2 | 52,9 (12–141) | 3/2/0 | 13 (4,7–33) | 4,3 (3,26–7,79) | 198 (95–212) | 0 | 1 | 1 |
| REL (n = 5) | | 3/2 | 52 (33–164) | 2/2/1 | 57,5 (1,1–109) | 10,88 (3,9–22,05) | 204 (52–420) | 0 | 2 | 2 |
| HD (n = 5) | | 3/2 | N/A | N/A | N/A | N/A | N/A | N/A | N/A | N/A |
| Tab 1B | sex | age | months from DG | BCR-ABL (%) | | WBC (10^9/l) | PLT (10^9/l) | BL (%) | TH | |
| NR | F | 49 | 0 | 18,4 | | 66,59 | 454 | 1 | | |
| | | | 11 | 4,7 | | 5,75 | 178 | 0 | imatinib | |
| OR | F | 67 | 0 | 52 | | 56,11 | 502 | 1 | | |
| | | | 11 | 0,001 | | 5,91 | 216 | 0 | imatinib | |

do not have BCR-ABL and therefore are technically not "TKI resistant", we designated them as such for clarity.

The origin of the cells was cytogenetically verified (for a detailed commentary of cytogenetic analysis, see S2 Fig in S1 File), and properties of the new cell lines were characterized as described below.

**Cell growth.** We assessed cell growth and viability by trypan blue staining after treatment with either 10 μM imatinib or 100 nM dasatinib. Compared to their parental cell lines, all resistant sub-lines exhibited decreased sensitivity to TKIs. Furthermore, various levels of cross-resistance (i.e. resistance to both TKIs used) were present in all resistant cells, while the growth of BCR-ABL negative cells remained unaffected by the presence of TKIs.

**Sensitivity to TKI.** Table 2 shows the EC50 values calculated from the measurement of cell activity after 48 h TKI treatment (S3 Fig in S1 File). All resistant sub-lines demonstrated lower sensitivity/higher EC50 values to both TKIs, and cross-resistance was evident in all sub-lines, similar to the results of cell growth experiments.

**Table 2. EC50 values of imatinib and dasatinib effects on the proliferation of sensitive and resistant cells.** EC50 values of imatinib and dasatinib on all used cells. Cell lines were incubated with 0–100 μM imatinib or 0–100 nM dasatinib. The EC50 on cell proliferation/viability was assessed by the Alamar Blue. EC50 values and CI95% were calculated from 2–3 independent experiments.

| | imatinib (μM) | dasatinib (nM) |
|---|---|---|
| JURL-MK1 | 0,18 (0,16–0,20) | 0,16 (0,14–0,19) |
| JURL-MK1 IR | 25,3 (17,2–38,1) | 35,7 (23,5–63,3) |
| JURL-MK1 DR | 0,7 (0,6–0,9) | 2,0 (1,5–2,6) |
| MOLM-7 | 0,25 (0,19–0,31) | 0,15 (0,11–0,19) |
| MOLM-7 IR | 59,2 (12,6–188,0) | 1,4 (0,7–2,6) |
| MOLM-7 DR | 92,7 (35,7–783,8) | 36,7 (26,3–55,7) |
| K562 | 0,25 (0,12–0,42) | 0,34 (0,25–0,47) |
| K562 IR | 4,8 (3,5–6,5) | 3,5 (1,8–6,8) |
| K562 DR | 5,2 (2,9–9,3) | 3,6 (1,7–7,5) |

**BCR-ABL transcript variants and mutations in the BCR-ABL kinase domain.**   To further characterize the novel cell lines, we verified the BCR-ABL transcript present in each cell type. The variant found in the parental line was always preserved in the derived resistant sublines. Specifically, MOLM-7 cells contained the e13a2 (b2a2) transcript, while JURL-MK1 and K562 contained the e14a2 (b3a2) transcript.

The T315I mutation in the kinase domain was present in both JURL-MK1 IR and DR cells, while we did not detect any mutations in the kinase domain of BCR-ABL in resistant cells derived from MOLM-7 and K562. Sequences are included as S1A-S1C Table in S1 File.

**BCR-ABL activity.**   We assessed BCR-ABL activity by evaluating the phosphorylation status of CRKL, a surrogate marker of BCR-ABL activity. In JURL-MK1 and MOLM-7 TKI-resistant sub-lines, P-CRKL levels were increased compared to parental cells. Interestingly, in K562 IR cells, CRKL phosphorylation was significantly decreased compared to their sensitive counterparts (Fig 1).

CK2 activity is enhanced in TKI resistant cell lines We also evaluated the levels of HSP90β P-S226 and CK2 in TKI-resistant cells to verify the results obtained from protein arrays (Fig 2, S4 Fig in S1 File). The level of phosphorylated HSP90β-S226 was significantly increased in both JURL-MK1 derived, TKI-resistant sub-lines compared to the sensitive counterparts. We therefore assessed the expression of CK2 subunits and evaluated CK2 activity using a specific antibody for the detection of CK2 phosphorylated substrates (Fig 2B).

While the CK2 α and β subunit levels were increased in both IR and DR JURL-MK1 cells (Fig 2A), the same could not be said for MOLM-7 resistants (S4 Fig in S1 File), and in K562 cells, the CK2 subunit expression was lower in resistant cells than in the parental cell line (S4B Fig in S1 File). In all cells, phosphorylation of CK2 substrates and CDC37 was correlated with the CK2 subunit levels (Fig 2B, S4A and S4B Fig in S1 File).

We used phosphorylation of CDC37 on S13, which is known to be mediated by CK2 and can be used as a marker of CK2 activity [28], in subsequent experiments to assess CK2 inhibition

## CK2 inhibition reduces cell proliferation and viability of TKI resistant cells

To inhibit CK2, we used CX-4945 (Silmitasertib) as an inhibitor of choice. Based on the literature [29] and our own preliminary experience, 10 μM CX-4945 was used in all subsequent

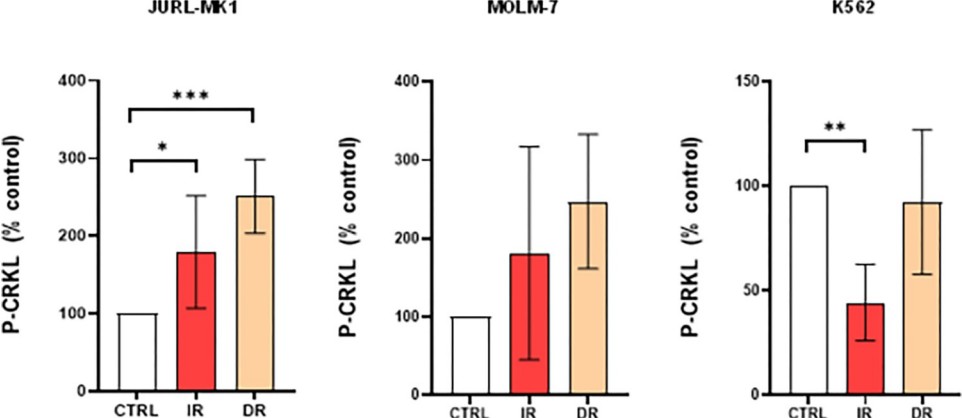

**Fig 1. BCR-ABL activity of TKI-resistant cells.** Relative densitometric graphs of phosphorylation of CRKL, surrogate marker of BCR-ABL activity. IR—imatinib resistant, DR—dasatinib resistant. Phosphorylation levels were normalised to β-actin and related to the corresponding controls. Means and standard deviation obtained from at least 3 biological replicates are shown. (***P< 0.001; **P<0.01; *P<0.05).

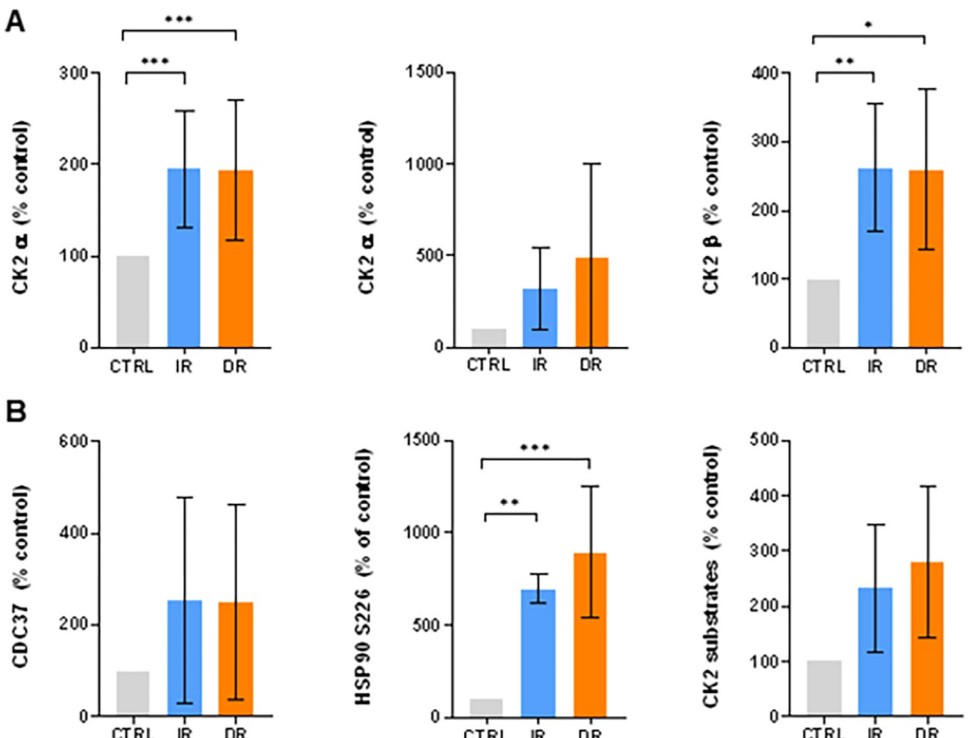

**Fig 2. CK2 subunits and CK2 substrate phosphorylation in JURL-MK1 cells.** Western blot analysis of JURL-MK1 cells and their resistant counterparts using antibodies against (A) CK2 subunits and (B) CK2 substrates CDC37, HSP90 S226, and general CK2 consensus substrate sequence (pS/pT)DXE. Protein/phosphorylation levels were normalized to β-actin and related to the corresponding controls. Means and standard deviation obtained from at least 3 biological replicates are shown. (***P < 0.001; **P<0.01; *P<0.05).

experiments. Proliferation ability of the cells was measured by Alamar Blue. Changes induced by TKIs and CX-4945 are shown in Fig 3A and 3B.

Fig 3A shows that CX-4945 decreased cell proliferation in all parental CML cells. However, the effect of TKIs alone was more significant. Although the newly acquired resistance did not affect the impact of CX-4945, the previously strong effects of TKIs were greatly reduced in cells with acquired resistance. Notably, the combination of CX-4945 with TKIs had more profound effect on the resistant cells than either inhibitor alone, indicating potentiation (with the exception of JURL-MK1 resistants). In BCR-ABL negative OCI-AML3, the effect of inhibitor combination was comparable to that of CX-4945 alone and was not affected by acquired resistance of the OCI-AML3 cells (S5A Fig in S1 File).

Similar results were observed for the cell count and dead cell fraction. Although TKIs were effective in JURL-MK1 and MOLM-7 sensitive cell lines, they failed to exhibit a significant effect on the resistant cells (Fig 3B). In K562 cells, the effect of TKIs alone was similar for the sensitive and resistant cells. This correlated with the percentage of growth inhibition assessed by alamar blue (Fig 3A). Once again, in BCR-ABL negative OCI-AML3 cells, CX-4945 treatment resulted in a slight growth inhibition (S5B Fig in S1 File), but CK2 inhibition did not increase the dead cell fraction in those cells.

In cases where TKIs or CX-4945 had taken effect on cell proliferation and viability, such as in BCR-ABL positive cells, we were detected PARP cleavage, indicating an apoptotic mechanism of cell death (Fig 3C, S6 Fig in S1 File).

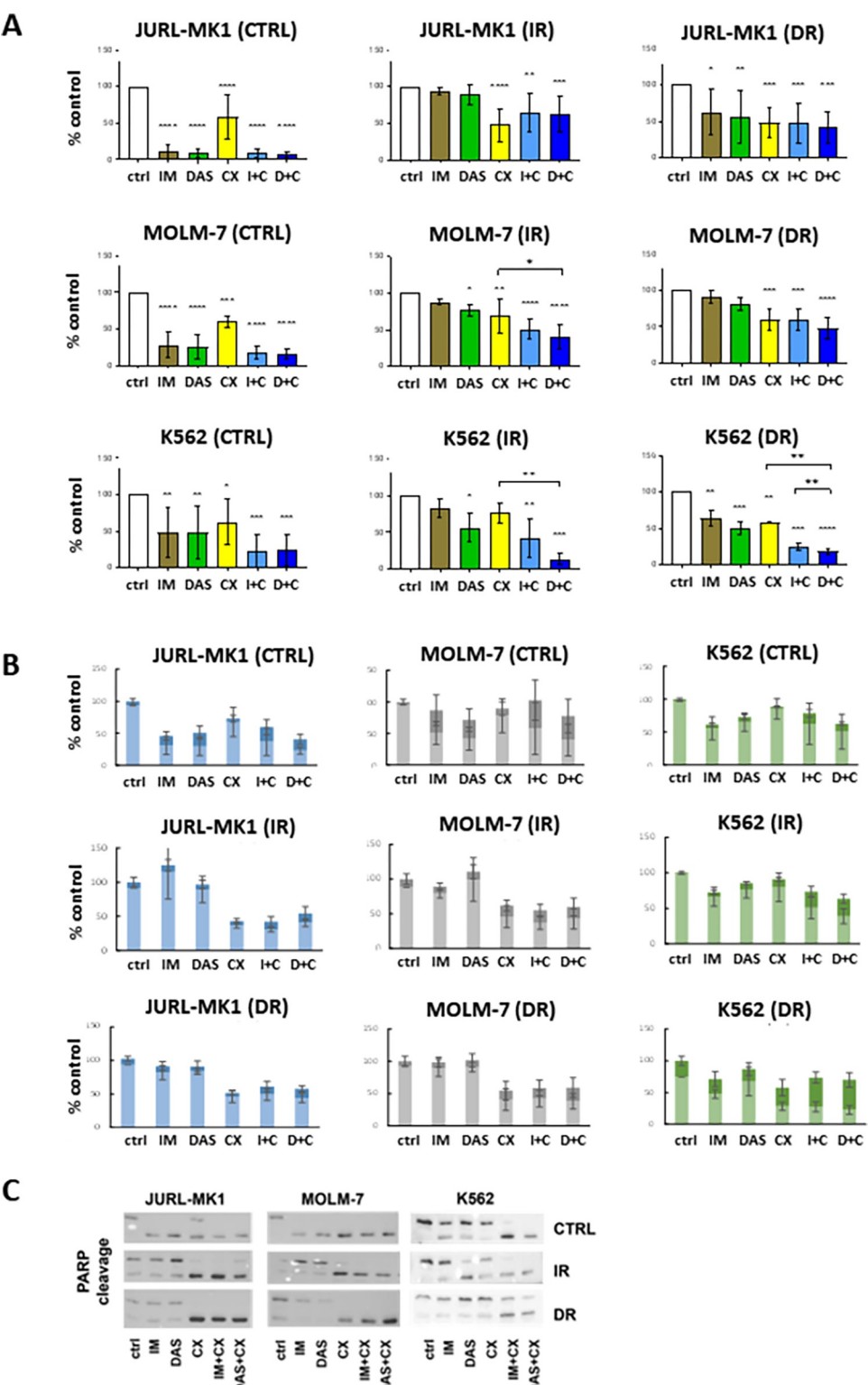

**Fig 3. Effects of CX-4945 on cell proliferation and viability.** The cells were treated for 48 h with imatinib (IM, 10 μM), dasatinib (DAS, 100 nM), CX-4945 (CX, 10 μM), and their combinations. (A) Proliferation activity of the cells was assessed by the Alamar blue method. The data represents mean and SD of 3–6 separate experiments. Statistical significance was assessed by one-way ANOVA followed by a Dunnett's multiple comparisons test. (B) Cell count and viability were evaluated by trypan blue staining. Light bars—live cells, dark bars—dead cells. (C) Apoptosis was tested

via cleaved PARP1 by western blots. Representative western blots are presented. Means and standard deviation obtained from at least 3 independent biological replicates are shown. ****$P<0.0001$, ***$P < 0.001$, **$P<0.01$, *$P<0.05$. C—control/naïve cells, IR—imatinib resistant, DR—dasatinib resistant.

## CK2 is regulated by BCR-ABL, and participates in the TKI resistance phenotype

There are conflicting views on the issue of how CK2 and BCR-ABL regulate each other [30,31]. Therefore, we aimed to assess the relationship between these kinases in our new cell models by evaluating changes in the phosphorylation of their prominent substrates, CDC37 and CRKL, after inhibitor treatment (Fig 4).

As predicted, TKIs had a significant effect on CRKL phosphorylation in sensitive cells, while CX-4945 did not significantly alter the P-CRKL levels, indicating that CK2 inhibition does not affect BCR-ABL activity. Phosphorylation of CDC37, on the other hand, was affected by the treatment with TKIs as well as with CX-4945, indicating that BCR-ABL activity is necessary for CK2 activation.

In resistant cells, CRKL phosphorylation in JURL-MK1 and MOLM-7 resistant cells remained largely unchanged upon inhibitor treatment. However, in K562 cells, TKI treatment led to a moderate decrease in CRKL levels. This may explain the slight reduction of proliferation and viability impairment observed upon TKI treatment in the K562 resistants (see Fig 3A). Phosphorylation of CDC37 in the resistant cells was largely unaffected by TKIs, the exception being JURL-MK1_IR cells, where CDC37 phosphorylation was significantly higher in TKI-treated cells. The same trend was observed in K562_IR cells, but was not statistically significant. In JURL-MK1 and MOLM-7 resistants, CX-4945 reduced CDC37 phosphorylation alone and in combination with the TKIs. In K562 resistant cells, the effect of combined CK2 and BCR-ABL inhibition was more significant than CK2 inhibition alone. This was statistically significant for K562 DR cells.

## CK2 inhibition decreases viability and induces apoptosis of primary CML cells

In order to quantify the efficacy of CK2 inhibition in primary CML cells, we obtained samples from two patients with CML who were unresponsive response to other TKIs (nilotinib and bosutinib), and who did not have mutations in the BCR-ABL kinase domain (patient characteristics are listed in Table 3).

We therefore incubated the cells with imatinib, dasatinib, and CX-4945 under the same conditions as the cell lines used in our previous experiments. Patient response to the TKIs varied, with patient 1 being more sensitive than patient 2 (Fig 5A). Patient 2's sensitivity was surprising, given the clinical failure of bosutinib (see to Table 3).

We also investigated the effects of a combination of TKIs and CX-4945 (Fig 5B). In patient 2 (who was unresponsive to TKIs), the combination of CX-4945 with TKIs significantly reduced the proliferation of patient cells, while having only a minor effect on healthy donor samples.

## Discussion

In our previous work, we demonstrated that increased HSP90 levels correlate with negative outcomes in chronic myeloid leukemia (CML) course and response to therapy [32]. Here, we identified increased HSP90β S226 phosphorylation in patients with poor response (either non-responding or relapsed) to treatment with tyrosine kinase inhibitors (TKIs, S1A Fig in S1 File).

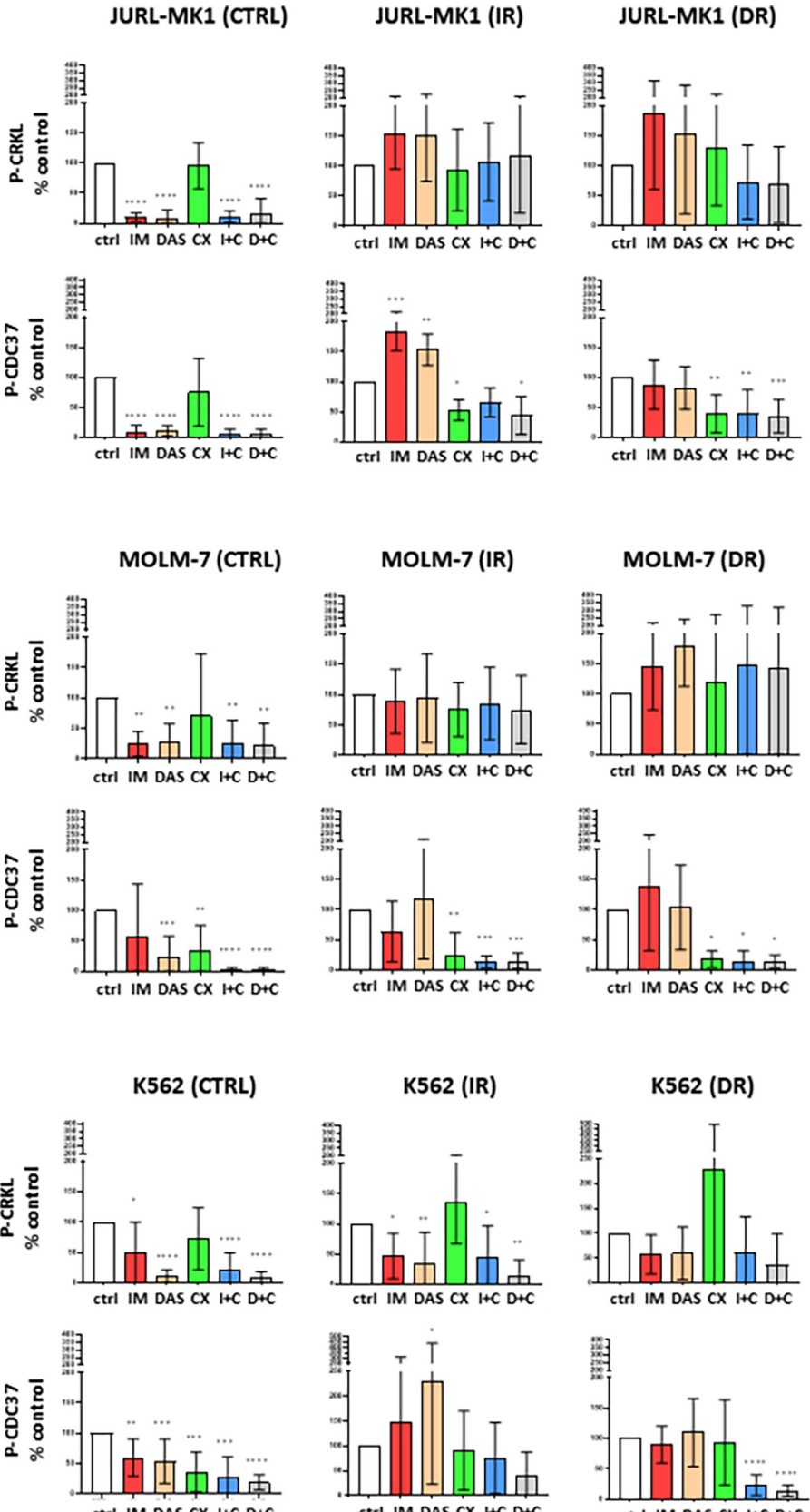

**Fig 4. Effect of CX-4945 and TKIs on CK2 and BCR-ABL activity.** The cells were treated for 48 h with imatinib (10 μM, IM), dasatinib (100 nM, DAS), CX-4945 (10 μM, CX), and their combinations. Quantification of western blot analyses of their effect to phosphorylation of CDC37 and CRKL are shown. Means and standard deviation obtained from at least 3 biological replicates. (A) JURL-MK1, (B) MOLM-7, (C) K562; ****$P < 0.0001$, ***$P < 0.001$, **$P < 0.01$, *$P < 0.05$.

Changes in HSP90β phosphorylation (both hyperphosphorylation and hypophosphorylation) are linked to TKI resistance in CML [33]. In this study, we observed a marked increase in the ratio of HSP90β P-S226 to total HSP90, even after one year of treatment, in a patient who did not respond to therapy (S1B Fig in S1 File).

HSP90 is a known substrate of CK2 [34–36]. Given the established role of CK2 in several cancers, we hypothesized that CK2 plays a role in CML resistance to TKIs. Since the availability of patient samples is limited, we generated six new, CML-derived cell lines that are resistant to TKIs. All the "resistant" cell lines were able to grow in media containing 2 μM imatinib or 2 nM dasatinib (Table 2, S3 Fig in S1 File). The observed cross-resistance (insensitivity of the cells to another TKI than that used for resistance acquisition) confirms that the resistance mechanisms developed under the selective pressure of one inhibitor often provide resistance to other TKIs as well [37].

To illustrate the differences among these cell lines, we characterized the cells based on their cytogenetic profile, BCR-ABL activity, and CK2-related protein content (S2 and S4 Figs in S1 File, Figs 1 and 2) to illustrate differences among these cell lines. Mutation of the BCR-ABL kinase domain is the main mechanism by which CML cells acquire resistance to TKIs. In cases where BCR-ABL mutations do not develop, BCR-ABL gene amplification is usually present [38,39]. In our study, JURL-MK1 resistant cells developed the T315I mutation in the BCR-ABL kinase domain but showed no additional amplification of the BCR-ABL fusion gene, while MOLM-7 and K562-derived resistants did not bear BCR-ABL mutation, but had amplified BCR-ABL gene (S2 Fig in S1 File). This led to a comparison of BCR-ABL activity in those cells (Fig 1), where the phosphorylation of a BCR-ABL activity surrogate marker, CRKL, was markedly increased in JURL-MK1 and MOLM-7. Interestingly, P-CRKL signal was significantly decreased in K562 IR cells compared to TKI-sensitive cells, despite the IR cells possessing additional (albeit non-amplified) BCR-ABL fusions in their genome (S2 Fig in S1 File). This is not unexpected, as others [40,41] have also observed amplification of BCR-ABL gene without any KD mutation in imatinib-resistant cell lines.

The expression changes of CK2 subunits were found to be variable across different cell lines, as illustrated in Fig 2 and S4 Fig in S1 File. Interestingly, in JURL-MK1 and K562 cells, the alteration in CK2 subunits correlated with the phosphorylation of CK2 substrates and CDC37, a surrogate marker for CK2 activity. In contrast, the expression of CK2 subunits in resistant MOLM-7 cells did not change significantly. However, we observed a trend of increased phosphorylation of CK2 substrates, which may be attributed–at least in part–to the interplay between CK2 and BCR-ABL (Fig 4).

**Table 3. Patient characteristics.** DG—diagnosis, WBC—total leukocyte count, PLT—platelet count, BL—blasts.

| | Sex | Age at DG | Months from DG | BCR-ABL (%) | WBC (10^9/l) | PLT (10^9/l) | BL (%) | TH | aberations |
|---|---|---|---|---|---|---|---|---|---|
| | | | | | | | | at the time of analysis | |
| **P1** | M | 49 | 199 | 30 | 11,29 | 171 | | bosutinib | |
| **P2** | M | 61 | 93 | 7,6 | 49,4 | 52 | | nilotinib | |
| | | | 102 | 151 | 6,05 | 132 | 7,4 | bosutinib | +8, der(22)t(9;22)(q34;q11) |

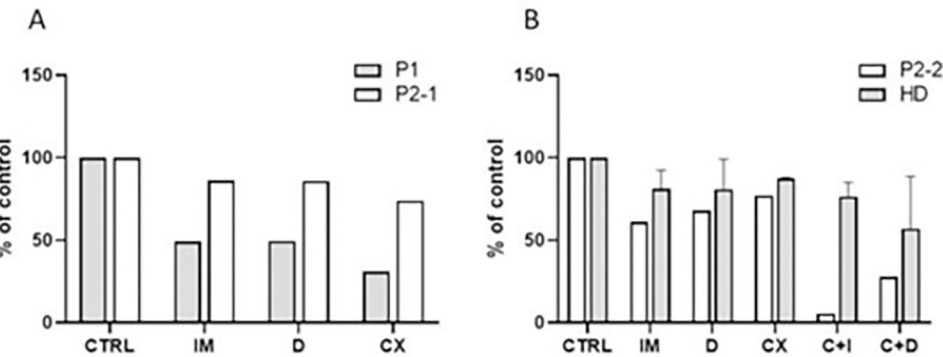

**Fig 5. Effect of CK2 inhibition on primary cells from CML patients.** (A) The cells collected from two CML patients non-responding to primary therapy were treated with imatinib (10 µM, IM), dasatinib (100 nM, D), or CX-4945 (10 µM, CX). Cell viability was assessed after 48 h incubation. (B) The cells from patient 2 (P2) were incubated with the inhibitors alone (IM, D, CX) or in combination (C+I, C+D). Viability after 48 h was compared with healthy donor.

In the context of CML, it is known that CK2 interacts with BCR-ABL [34,35], and imatinib-resistant cells show sensitivity to CK2 inhibitor CX-4945 [22,26]. This is also reflected in our cell models (Fig 3A and 3B). Our study also revealed a similar effect of CX-4945 in dasatinib-resistant cells, resulting in reduced cell viability and induction of cell death (Fig 3, S5 and S6 Figs in S1 File). Notably, BCR-ABL signaling remained active in all cells even after CK2 inhibition (Fig 4). Interestingly, CK2 activity in sensitive cells was affected by targeting BCR-ABL with imatinib or dasatinib, whereas this was not the case in resistant cells (Fig 4). Moreover, in cells with high BCR-ABL activity (i.e. JURL-MK1 and MOLM-7 resistants, as seen in Fig 1) the activity of CK2 correlated with that of BCR-ABL. However, in K562 resistants, the activity of CK2 remained the same or even increased despite suppression of BCR-ABL. This suggests that in resistant cells, CK2 activity is dependent on that of BCR-ABL, and that uncoupling of BCR-ABL and CK2 mutual regulation may be a part of the resistant phenotype. In support of this hypothesis, K562 resistant cells appeared to be the least dependent on BCR-ABL compared to JURL-MK1 and MOLM-7 resistants, and may rely on other candidate molecules such as Src family kinases (SFK), which are known to play a crucial role in TKI resistance [42,43], and are also regulated by (and regulate) CK2 [44,45].

Importantly, the combination of TKI and CX-4945 remained the most effective on viability and proliferation parameters in all cases (Fig 3), regardless of BCR-ABL activity (Fig 4). This suggests that CK2 inhibition may be crucial in overcoming CML resistance to TKIs, as the effects on cell proliferation and the increase of apoptotic parameters (Fig 3C) support the CK2 "addiction" hypothesis [17] in CML cells as well.

Moreover, as we and others have previously demonstrated [46–48], *in vitro/ex vivo* cultivation of isolated patient leukocytes with TKIs can serve as a reliable predictor of therapy results and have a significant prognostic value. Similarly, *in vitro* experiments with CK2 inhibitors may help determine the role of CK2 in resistance to TKIs, thus offering a potential avenue for targeting CK2 kinase. Notably, CK2 inhibition did not affect leukocytes from healthy donors (Fig 5), as previously reported [28]. Conversely, CK2 inhibition alone or in combination with TKI reduced the metabolic activity of primary cells from CML patients with resistance to various TKI (Fig 5).

In summary, our study provides insights into the effects of CK2 inhibition on TKI-resistant CML cells. By using six originally developed cell lines and primary patient samples, we demonstrate that CK2 inhibition and/or its combination with TKIs can induce cell death in TKI-resistant cells, including those bearing the gatekeeper T315I mutation. Importantly, BCR-ABL negative cell lines and primary cells from healthy donors remain unaffected by such treatment.

## Material and methods

### Patient samples and cell cultures

Leukocytes from CML patients and healthy donors were isolated as described previously [32,48]. Responses to therapy were defined according to European Leukemia Net (ELN) classification [23]. Samples were obtained with the agreement of the Ethics Committee of the Institute of Hematology and Blood Transfusion (Prague, Czech Republic) according to the Declaration of Helsinki. Written informed consent was obtained from all patients.

Cell lines were purchased from DSMZ (Braunschweig, Germany–OCI-AML3, JURL-MK1) or ECACC (Salisbury, UK–HL-60, K562). The cell line MOLM-7 (not commercially available) was obtained from Dr J. Minowada. From all used cell lines, sub-lines resistant either to 2 μM imatinib or 2 nM dasatinib were raised according to the protocol by Mahon et al. [49]. Briefly, resistant cells were developed by prolonged exposures to gradually increasing concentrations of imatinib and dasatinib, starting from 1 nM to 2 μM (imatinib) and 1 pM to 2 nM (dasatinib). Parental cell lines were cultivated in parallel without inhibitors. All cells were incubated at 37˚C in RPMI medium with 10% fetal bovine serum and antibiotics.

### Chemicals and antibodies

Imatinib mesylate, nilotinib, dasatinib, and CX-4945 were purchased from Santa Cruz Biotechnology, Inc. All inhibitors were dissolved as 10 mM stock solution in DMSO (Sigma-Aldrich Inc., Missouri, USA) and stored at -20˚C. Protease inhibitor cocktails and Phosphatase inhibitor cocktails were purchased from Calbiochem and SIGMA-ALDRICH, respectively. Non-conjugated primary antibodies casein kinase IIα (D-10 and E-7), casein kinase IIα´ Antibody (E-7), casein kinase IIβ Antibody (6D5), and Actin (anti-actin antibody clone AC-15, SIGMA-ALDRICH) or beta-tubulin (β-Tubulin (9F3) were purchased from SantaCruz Biotechnology (CA, USA). Anti-HSP90 beta (phospho S226) antibody was purchased from Abcam (UK). Phospho-CK2 Substrate [(pS/pT)DXE] MultiMab™ Rabbit mAb, Phospho-CDC37 (Ser13) (D8P8F) Rabbit mAb, Phospho-CrkL (Tyr207) Rabbit mAb, Cleaved PARP (Asp214) (D64E10) XP, and HRP conjugated secondary antibodies were purchased from Cell Signaling Technology (Danvers, MA, USA).

### Cell viability and proliferation

Cell viability was assessed by Trypan blue staining (according to the manufacturer's protocol); cell metabolic activity was evaluated using the AlamarBlue® assay (Invitrogen), following the manufacturer´s instructions. Briefly, 100 μL of cell suspension was transferred to a 96-well plate and 10 μL of AlamarBlue reagent was added to each well. After incubation at 37˚C for 1 hour, fluorescence intensity was measured on microplate reader BMG FLUOstar Galaxy (MTX Lab Systems, Inc., VA, USA). Statistical analysis was performed using GraphPad Prism 8.2.

### Immunoblotting

Western blot analyses were performed as described previously 68. Briefly, cells were harvested, lysed in Laemmli buffer, and boiled for 10 minutes. The samples were resolved on SDS polyacrylamide gel (10%), transferred to PVDF membrane, and incubated with the appropriate antibodies. Protein bands were detected by chemiluminescence (SuperSignal West Dura Extended Duration Substrate -Thermo Fisher Scientific, USA) and scanned using a G:BOX imager (Syngene Europe). Densitometric quantification was performed using Gene Tools product version: 4.3.8.0 (Syngene Europe). All analyses were repeated 2–5 times.

## BCR-ABL transcript variants analysis

**RT-PCR.** Total RNA was extracted by the NucleoSpin® RNA Plus Kit (Macherey-Nagel, Germany) according to the appropriate protocol. The cDNA synthesis was prepared using oligoT primers and SuperScript II transcriptase (Invitrogen/Life Technologies, USA). The cDNA of CML leukemic cell lines and control cell lines was used during the development. PCR was performed using a CFX96 Touch Real-Time PCR Detection Systems (Bio-Rad). Primer sequence for a2b2 (e13b2) and a2b3 (e14b2) transcript variants:

a2b2 (310 bp) BCR-b2 5´- ACAGAATTCCGCTGACCATCAATAAG-3´
ABL-a2 5´- TGTTGACTGGCGTGATGTAGTTGCTTGG-3´
a2b3 (385 bp) BCR-b2 5´- ACAGAATTCCGCTGACCATCAATAAG-3´
ABL-a2 5´- TGTTGACTGGCGTGATGTAGTTGCTTGG-3´

**Nested PCR.** Long-range nested RT-PCR analysis was performed to amplify BCR-ABL cDNA. For this purpose the sequences of primers used for RT PCR were as follows:

Step 1:
BCR1-F 5´- TGACCAACTCGTGTGTGAAACTC-3´
ABL1-R 5´-TCCACTTCGTCTGAGATACTGGATT-3´
Step 2:
ABL1-F 5´-CGCAACAAGCCCACTGTCT -3´
ABL1-R 5´-TCCACTTCGTCTGAGATACTGGATT-3´

Step 1: initial denaturation at 95˚C / 2 min, 40 cycles at 95˚C / 1 min, 60˚C / 1 min, and 72˚C / 3 min. Final 10-min extension step at 72˚C. Step 2: initial denaturation at 95˚C / 2 min, 50 cycles at 95˚C / 1 min, 60˚C / 1 min, and 72˚C / 3 min. Final 10-min extension step at 72˚C.

## Mutational analysis

PCR products were separated on 2% agarose gel containing MIDORIGreen Advance. Appropriate bands were cut out and purified with the QIAquick Gel Extraction kit (Qiagen). Sanger sequencing was performed on ABI PRISM 3500 Genetic Analyzer using Big Dye Terminator 3.1 kit (Applied Biosystems). The resulting sequences were analyzed in Chromas 2.31 program and tools/program of Blast was used for alignment of sequences.

## Cytogenetics

Cells were treated with demecolcin for 1,5 hours. Harvesting and preparation of slides were performed according to standard cytogenetic procedures. Cells were stored at -20˚C in methanol-glacial acetic acid (3:1). For cytogenetic analyses, cell suspensions were dropped on microscopic slides and air-dried. Fluorescence in situ hybridization (FISH) in combination with multicolor fluorescence in situ hybridization (mFISH) were used to characterize the chromosomes. Analyses were performed using commercially available probes BCR/ABL Vysis LSI BCR/ABL Dual Color, Dual Fusion Translocation Probe (Abbott Vysis, USA), and 24 XCyte mFISH Kit (MetaSystems, Germany). All available mitoses for both probes and two hundred nuclei for the LSI BCR/ABL probe were analysed using an AxioImager Z1 fluorescence microscope (Carl Zeiss, Germany) and the Isis computer analysis system (MetaSystems, Germany). Findings were described according to ISCN 2016 [50].

## Phosphorylation antibody microarray

The Phospho Explorer Antibody Array PEX100 containing 1,318 well-characterized site-specific antibodies was purchased from Full Moon Biosystems (Sunnyvale, CA, USA). The isolated whole fraction of leukocytes was processed according to the manufacturer's instructions.

Briefly: cells were lysed, labeled by Biotin, placed to glass array, and incubated overnight at 4˚C. Cy3-Streptavidin was used for protein visualization, signal to noise ratio was measured on GenePix® Microarray Scanner (4000A; Molecular Devices, USA).

## Statistical analysis

All statistical analysis was performed using GraphPad Prism 8.2 software (GraphPad Software, Inc.). Analyses were assessed by repeated-measures ANOVA, Dunnett's multiple comparisons test was used. Statistically significant results were obtained in independent biological replicates. $P < 0.05$ was considered statistically significant. Experiments were repeated at least three times. All data are presented as the mean ± standard deviation.

## Supporting information

**S1 File.**
(PDF)

**S1 Raw images.**
(PDF)

## Author Contributions

**Conceptualization:** Tereza Macháčková-Lopotová, Markéta Žáčková.

**Data curation:** Ondřej Mitrovský, Tereza Macháčková-Lopotová, Markéta Žáčková.

**Funding acquisition:** Markéta Žáčková.

**Investigation:** Ondřej Mitrovský, Denisa Myslivcová, Tereza Macháčková-Lopotová, Adam Obr, Kamila Čermáková, Šárka Ransdorfová, Jana Březinová, Markéta Žáčková.

**Methodology:** Ondřej Mitrovský, Denisa Myslivcová, Tereza Macháčková-Lopotová, Markéta Žáčková.

**Resources:** Hana Klamová.

**Supervision:** Markéta Žáčková.

**Writing – original draft:** Adam Obr, Markéta Žáčková.

**Writing – review & editing:** Ondřej Mitrovský, Denisa Myslivcová, Tereza Macháčková-Lopotová, Adam Obr, Markéta Žáčková.

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
