## [Decision Letter · Decision Letter 0]

24 Jan 2023

PONE-D-22-27194Inhibition of Casein Kinase 2 induces cell death in chronic myelogenous leukemia cells resistant to tyrosine kinase inhibitorsPLOS ONE

Dear Dr. Obr,

Thank you for submitting your manuscript to PLOS ONE. After careful consideration, we feel that it has merit but does not fully meet PLOS ONE’s publication criteria as it currently stands. Therefore, we invite you to submit a revised version of the manuscript that addresses the points raised during the review process.

We look forward to receiving your revised manuscript.

Kind regards,

Rama Krishna Kancha

Academic Editor

PLOS ONE

Journal Requirements:

3. "In your Data Availability statement, you have not specified where the minimal data set underlying the results described in your manuscript can be found. PLOS defines a study's minimal data set as the underlying data used to reach the conclusions drawn in the manuscript and any additional data required to replicate the reported study findings in their entirety. All PLOS journals require that the minimal data set be made fully available. For more information about our data policy, please see http://journals.plos.org/plosone/s/data-availability.

We will update your Data Availability statement to reflect the information you provide in your cover letter."

Additional Editor Comments (if provided):

While the reviewers acknowledge the value of this work, they have certain major concerns regarding the mutation status in leukemic cell lines used in this study. It is important to address questions raised by reviewers in a point-wise rebuttal.

Reviewers' comments:

Reviewer's Responses to Questions

**Comments to the Author**

1. Is the manuscript technically sound, and do the data support the conclusions?

Reviewer #1: No

Reviewer #2: Yes

2. Has the statistical analysis been performed appropriately and rigorously? 

Reviewer #1: Yes

Reviewer #2: Yes

3. Have the authors made all data underlying the findings in their manuscript fully available?

Reviewer #1: No

Reviewer #2: Yes

4. Is the manuscript presented in an intelligible fashion and written in standard English?

Reviewer #1: Yes

Reviewer #2: Yes

5. Review Comments to the Author

Reviewer #1: The manuscript entitled “Inhibition of Casein Kinase 2 induces cell death in chronic myelogenous leukemia cells resistant to tyrosine kinase inhibitors” by Adam Obr is interesting however the manuscript has many flaws which needed to be answered before its publication.

Throughout the manuscript the authors used sentences like “somewhat affected” and “slight growth slowing”. These sentences need to be avoided and “percentage of growth inhibition” would be recommended.

Instead of denoting K562 cells which have never been exposed to inhibitors as control, representation of them as näive K562 cells would be more appropriate.

Results part needs to be written very precisely and the sub heading should have a title which explains the outcome of the results otherwise it is highly confusing what authors want to claim.

When there is no explanation for CK2 levels and chromosomal abnormality devoting one big paragraph for cytogenetics is unnecessary and this part is highly confusing and should move to supplementary information.

Most importantly, authors should check the mutational status of BCR-ABL in the established TKI resistant cell lines. Without this information, concluding the results part is highly indecisive. Best example is Figure 1, TKI resistant JURL-MK1 and MOLM-7 cells showed persistent p-CRKL, whereas K562 could not show suggesting that in K562 cells TKI resistance might be due to BCR-ABL independent mechanism.

When data is not significant, showing them in the main figures is also confusing. Keeping them in the supplementary information and giving detailed explanation would be advisable. Example is Figure 2 CDC37 levels.

It is also highly recommended to authors to show the dose dependent growth curves in order to claim the specificity of the CX-4945 in the mentioned cell lines.

Apoptosis assays such as annexin stating needed to perform with different doses of CX-4945 showing the PARP cleavage is not enough. However, PARP cleavage was also noticeable in OCI-AML 3 cells with dasatinib and CX-4945 treatment. Do they get affected by TKIs treatment?

Authors claimed that OCI-AML 3 cell line is BCR-ABL negative and in Figure 3 C it is mentioned as imatinib resistant (IR) and Dasatinib resistant (DR) OCI-AML3, this is confusing, did they develop OCI-AML resistant clones against the imatinib and dasatinib? Authors should clearly check this information.

As said earlier, concluding the Figure 4 and relationship between the BCR-ABL and CK2 is indecisive without knowing the mutational status of BCR-ABL and mechanism of resistance in these clones.

Based on these points, the manuscript required major revision for publication.

Reviewer #2: This is a very nice study that can be accepted for publication with minor modifications and suggestions. The authors used proper analyses techniques and evaluate the results properly.

1. Page 7, Line 122 : T315I Mutation in BCR-ABL..

It is written as’’ In both JURL-MK1 IR and DR cells, there was a T315I mutation in the kinase domain. In resistant cells 123 derived from MOLM-7 and K562, respectively, no mutation in the kinase domain of BCR-ABL was 124 detected.’’in the results section of the manuscript. However, no DNA sequence data related to this mutation were included in this article. I think this result should be added to the article.

2. Please: consider adding a paper I wrote below to your references.

• ’’The casein kinase 2 inhibitor, CX-4945, as an anti-cancer drug in treatment of human hematological malignancies.’’

6. PLOS authors have the option to publish the peer review history of their article (what does this mean?). If published, this will include your full peer review and any attached files.

Reviewer #1: No

Reviewer #2: No

---

## [Author Response · Author response to Decision Letter 0]

5 Apr 2023

We thank both reviewers for their valuable input to our work.

Their specific points were addressed as follows.

Reviewer #1: 

The manuscript entitled “Inhibition of Casein Kinase 2 induces cell death in chronic myelogenous leukemia cells resistant to tyrosine kinase inhibitors” by Adam Obr is interesting however the manuscript has many flaws which needed to be answered before its publication.

Throughout the manuscript the authors used sentences like “somewhat affected” and “slight growth slowing”. These sentences need to be avoided and “percentage of growth inhibition” would be recommended.

- We changed the language used in the manuscript in order to be more precise (see the version with tracked changes). The results & discussion sections were largely rewritten for clarity and precision.

Instead of denoting K562 cells which have never been exposed to inhibitors as control, representation of them as näive K562 cells would be more appropriate.

- The part is edited now (line 167 in the revised version).

Results part needs to be written very precisely and the sub heading should have a title which explains the outcome of the results otherwise it is highly confusing what authors want to claim.

- The headings in the results section were updated.

When there is no explanation for CK2 levels and chromosomal abnormality devoting one big paragraph for cytogenetics is unnecessary and this part is highly confusing and should move to supplementary information.

- The part served mainly as a verification of the novel cell line origin and as a method for seeking BCR-ABL amplifications, which could be potentially relevant. 

To avoid confusion with detailed description, we largely redacted this part and removed is as separate section. The reference in the main text is now included at the beginning of the results section (lines 100-102 in the revised version, and referenced in other places) and the original, detailed commentary is moved to the supplementary material. 

Most importantly, authors should check the mutational status of BCR-ABL in the established TKI resistant cell lines. Without this information, concluding the results part is highly indecisive. Best example is Figure 1, TKI resistant JURL-MK1 and MOLM-7 cells showed persistent p-CRKL, whereas K562 could not show suggesting that in K562 cells TKI resistance might be due to BCR-ABL independent mechanism.

- We did check the mutation status in the cell lines. The sequences are now included as the Supplementary Table 1. The commentary was present in the original manuscript, and is also present in the revised version (lines 124-126). 

When data is not significant, showing them in the main figures is also confusing. Keeping them in the supplementary information and giving detailed explanation would be advisable. Example is Figure 2 CDC37 levels.

- CDC37 levels in figure 2, though statistically insignificant, show a clear trend. We feel that albeit not significant in this particular figure, without the results the manuscript would be more difficult to follow through, because we use CDC37 as a surrogate marker of CK2 activity. 

We did our best to reduce the main results to absolute minimum we feel is necessary to comprehend the main point (for example, we eliminated the PARP cleavage in OCI-AML3 from Fig 3C, see below). 

It is also highly recommended to authors to show the dose dependent growth curves in order to claim the specificity of the CX-4945 in the mentioned cell lines.

- The concentration of CX-4945 was chosen based on several publications and our own preliminary data. Since CX-4945 is a clinically tested inhibitor and therefore proven highly specific at given concentrations, addition of more results would not benefit the comprehensibility of the manuscript.

Apoptosis assays such as annexin stating needed to perform with different doses of CX-4945 showing the PARP cleavage is not enough. However, PARP cleavage was also noticeable in OCI-AML 3 cells with dasatinib and CX-4945 treatment. Do they get affected by TKIs treatment?

- This is actually a very interesting phenomena. In our lab experience, OCI-AML 3 cells show weak PARP cleavage even when they are sustained in a culture, untreated, with good proliferation and viability markers. You can see this in Fig 3C of the original manuscript in the control OCI-AML3. We currently have no explanation of this phenomena. 

For the sake of this manuscript, we would argue that the changes in PARP cleavage in OCI-AML3 cells are not significant (see quantification in Supplementary Fig 6). However, seeing that it might cause some confusion, we removed OCI-AML3 cells from the main Fig 3C, and kept the data only as a quantification in Supplementary Fig 6).

Authors claimed that OCI-AML 3 cell line is BCR-ABL negative and in Figure 3 C it is mentioned as imatinib resistant (IR) and Dasatinib resistant (DR) OCI-AML3, this is confusing, did they develop OCI-AML resistant clones against the imatinib and dasatinib? Authors should clearly check this information.

- BCR-ABL negative cells are technically not TKI resistant (TKIs have no significant effect on them, since there is no BCR-ABL proliferation and cell survival dependence). Despite this, we did cultivate these cells in the same conditions as the new BCR-ABL positive cells (that means, in 2 µM imatinib or 2 nM dasatinib), and we labeled cells cultivated in those conditions as „TKI resistant“ for the sake of simplicity.

We now clarified this information in the results section (lines 97-99 in the revised version). 

As said earlier, concluding the Figure 4 and relationship between the BCR-ABL and CK2 is indecisive without knowing the mutational status of BCR-ABL and mechanism of resistance in these clones.

Based on these points, the manuscript required major revision for publication.

- Hopefully, we addressed your points well enough. Thank you for the comments.

Reviewer #2: 

This is a very nice study that can be accepted for publication with minor modifications and suggestions. The authors used proper analyses techniques and evaluate the results properly.

1. Page 7, Line 122 : T315I Mutation in BCR-ABL..

It is written as’’ In both JURL-MK1 IR and DR cells, there was a T315I mutation in the kinase domain. In resistant cells 123 derived from MOLM-7 and K562, respectively, no mutation in the kinase domain of BCR-ABL was 124 detected.’’in the results section of the manuscript. However, no DNA sequence data related to this mutation were included in this article. I think this result should be added to the article.

- The sequences related to line 122 in the original manuscript are now included as Supplementary Table 1. The commentary is also present in the revised version, lines 124-126.

2. Please: consider adding a paper I wrote below to your references.

• ’’The casein kinase 2 inhibitor, CX-4945, as an anti-cancer drug in treatment of human hematological malignancies.’’

- The reference was added, thank you for the suggestion.

Thank you for your kind comments, hopoefully we addressed them well enough.

---

## [Editor Report · Decision Letter 1]

11 Apr 2023

Inhibition of Casein Kinase 2 induces cell death in chronic myelogenous leukemia cells resistant to tyrosine kinase inhibitors

PONE-D-22-27194R1

Dear Dr. Obr,

We’re pleased to inform you that your manuscript has been judged scientifically suitable for publication and will be formally accepted for publication once it meets all outstanding technical requirements.

Kind regards,

Rama Krishna Kancha

Academic Editor

PLOS ONE
---

## [Editor Report · Acceptance letter]

18 Apr 2023

PONE-D-22-27194R1 

Inhibition of Casein Kinase 2 induces cell death in tyrosine kinase inhibitor resistant chronic myelogenous leukemia cells 

Dear Dr. Obr:

I'm pleased to inform you that your manuscript has been deemed suitable for publication in PLOS ONE. Congratulations! Your manuscript is now with our production department. 

Kind regards, 

on behalf of

Dr. Rama Krishna Kancha 

Academic Editor

PLOS ONE